# Role of 3D printing technology in paediatric teaching and training: a systematic review

Ashar Asif ,[1] Elgin Lee ,[2] Massimo Caputo ,[1,3] Giovanni Biglino ,[3,4] Andrew Ian Underwood Shearn [1,3]

[1]Bristol Medical School, University of Bristol, Bristol, UK
[2]Children's Services Directorate, Newcastle Upon Tyne Hospitals NHS Foundation Trust, Newcastle Upon Tyne, UK
[3]Bristol Heart Institute, University Hospitals Bristol and Weston NHS Trust, Bristol, UK
[4]National Heart and Lung Institute, Imperial College London, London, UK

**Correspondence to**
Dr Andrew Ian Underwood Shearn; andrew.shearn@bristol.ac.uk

## ABSTRACT

**Background** In the UK, undergraduate paediatric training is brief, resulting in trainees with a lower paediatric knowledge base compared with other aspects of medicine. With congenital conditions being successfully treated at childhood, adult clinicians encounter and will need to understand these complex pathologies. Patient-specific 3D printed (3DP) models have been used in clinical training, especially for rarer, complex conditions. We perform a systematic review to evaluate the evidence base in using 3DP models to train paediatricians, surgeons, medical students and nurses.

**Methods** Online databases PubMed, Web of Science and Embase were searched between January 2010 and April 2020 using search terms relevant to "paediatrics", "education", "training" and "3D printing". Participants were medical students, postgraduate trainees or clinical staff. Comparative studies (patient-specific 3DP models vs traditional teaching methods) and non-comparative studies were included. Outcomes gauged objective and subjective measures: test scores, time taken to complete tasks, self-reported confidence and personal preferences on 3DP models. If reported, the cost of and time taken to produce the models were noted.

**Results** From 587 results, 15 studies fit the criteria of the review protocol, with 5/15 being randomised controlled studies and 10/15 focussing on cardiovascular conditions. Participants using 3DP models demonstrated improved test scores and faster times to complete procedures and identify anatomical landmarks compared with traditional teaching methods (2D diagrams, lectures, videos and supervised clinical events). User feedback was positive, reporting greater user self-confidence in understanding concepts with users wishing for integrated use of 3DP in regular teaching. Four studies reported the costs and times of production, which varied depending on model complexity and printer. 3DP models were cheaper than 'off-the-shelf' models available on the market and had the benefit of using real-world pathologies. These mostly non-randomised and single-centred studies did not address bias or report long-term or clinically translatable outcomes.

**Conclusions** 3DP models were associated with greater user satisfaction and good short-term educational outcomes, with low-quality evidence. Multicentred, randomised studies with long-term follow-up and clinically assessed outcomes are needed to fully assess their benefits in this setting.

**PROSPERO registration number** CRD42020179656.

### What is known about the subject?

► Undergraduate paediatric education is known to be short and postgraduate paediatric training can be limited within certain regions.
► Patients with congenital diseases are surviving to adulthood, meaning adult clinicians are seeing an increase of cases which were traditionally considered as 'paediatric conditions'.
► Patient-specific 3D printed (3DP) models have been used in other medical and surgical specialties for hands-on education and training and can compensate for these pitfalls.

### What this study adds?

► 3DP models in paediatric education and training sessions have been shown to improve immediate educational and procedural performance, with high user satisfaction.
► The quality of evidence is poor due to unaddressed confounding factors, small study cohorts and poor study design.
► In the few studies that reported costs, 3DP models had the benefit of producing models at relatively lower costs compared with alternative resources.

## INTRODUCTION

In 2017, the Royal College of Paediatrics and Child Health published a response to the 'Shape of Training Review' of postgraduate training within the UK. They observed that 'doctors coming into paediatric specialty training do so from a lower knowledge base than adult medicine'.[1] This could partly be explained by the brevity of undergraduate exposure to paediatrics, typically 4–8 weeks during the entirety of medical school.[2] Furthermore, some paediatric trainees report a lack of educational opportunities within their postgraduate training programme,[3] potentially driven by the varied facilities offered by different regions and centres.[4 5]

It is increasingly important for non-paediatric clinicians to fully appreciate paediatric pathologies. Novel therapies have allowed many children, who would have had poor outcomes from previously life-limiting diseases, to now live with such pathologies resulting in complex patients with chronic conditions.[5] Clinical care and observation over the adult lifetime of these patients is often overseen by a myriad of clinicians, frequently requiring specialised oversight such as in the case of pregnant women with congenital heart disease (CHD).[6]

In recent years, 3D printing has played a role in a variety of medical and surgical training and educational settings.[7–9] 3D printed (3DP) models can replicate in vivo clinical pathologies based on radiological images, allowing trainees to be exposed to pathologies and potential procedures they would not ordinarily experience through their routine clinical exposure.[10] With this review, we aimed to evaluate the current evidence base regarding the utility of 3DP models compared with traditional teaching methods (ie, 2D images/diagrams, lectures, videos and ward-based teaching) in educating and training of medical students and clinical trainees in paediatrics.

## METHODS
### Study protocol
The study protocol is registered on International Prospective Register of Systematic Reviews and can be found with the following ID: CRD42020179656. Throughout the review, the authors referred and adhered to the Preferred Reporting Items for Systematic Reviews and Meta-Analyses (PRISMA) guidelines.[11]

### Literature search strategy
The databases PubMed, Web of Science and Embase were searched. The search included all published material since January 2010 (last accessed in April 2020) in the English language. This time frame was selected to ensure the studies reviewed used the most recent available technology. The following terms were used across all databases: ((3d-prin*) OR (3d prin*) OR (three-dimensional print*) OR (three dimensional print*) OR (3-d print*)) AND ((paediat*) OR (pediat*)) AND ((education) OR (teaching) OR (training) OR (session) OR (simulation)). Synonyms and spelling variations were also searched to allow for international spellings and alternative terms (eg, paediatrics vs paediatrics). Searches on PubMed included Medical Subject Headings.

### Selection criteria
The inclusion criteria for final review were studies using 3DP models derived from paediatric patient imaging data (eg, CT, MRI, ultrasound or echocardiography) and that occurred in an educational or training setting. This was defined as a preplanned session or programme as part of an educational syllabus, training programme or professional course separate from a clinical setting and not aimed to directly contribute towards patient outcomes. This excluded studies which used 3DP models to facilitate communication with patients, carers or clinical staff. Studies that produced a 3DP model for preoperative planning for a specific case were also excluded. Included studies evaluated the use of patient-specific models with physicians, surgeons, nurses and allied health professionals undergoing teaching, training or simulation. Comparative studies would compare the use of 3DP models against traditional educational methods such as lectures, tutorials, practical sessions or textbooks. Case reports and case series were also included if they used the models in an educational setting and reported educational outcome measures. Published conference abstracts were included as per PRISMA guidance to minimise publication bias.[12] Primary outcome measures were either objective measures such as preintervention/postintervention testing scores or procedure performance, or subjective measures such as participant-reported opinions from questionnaires. Additional outcome measures that were recorded included the time taken to produce models and the cost of model production.

### Data extraction and appraisal of evidence
The titles and abstracts of all papers obtained during the searches were screened for relevance by two authors independently (AA and EL). Relevant papers were then independently assessed for eligibility in full against the study protocol (AA and EL). Any disagreements were resolved by consensus, with the senior author (AS) acting as the final adjudicator for any unresolved discrepancies. The final included papers were then reviewed and appraised by AA and EL. Data were extracted manually onto a Microsoft Excel (V2110, Microsoft, Redmond, WA, USA) spreadsheet.

### Patient and public involvement
No patients were involved, nor were any patient data used, in the development or analysis of this review.

## RESULTS
### Literature search outcomes
Our search resulted in a total of 587 articles. The screening for eligibility is illustrated in figure 1. A final total of 15 articles were included, with no additional articles identified from cross-referencing.

### Characteristics of selected studies
The studies included in this review focused on five main specialties relating to paediatric or congenital disorders: cardiovascular surgery, general surgery (one study shared with urology), neurosurgery, respiratory medicine and gynaecology. Figure 2 breaks down the included studies by subspecialty.

The majority of eligible studies (n=10) assessed the use of 3DP models in CHD. Out of the 15 included studies, 3 assessed the utility of 3DP models among

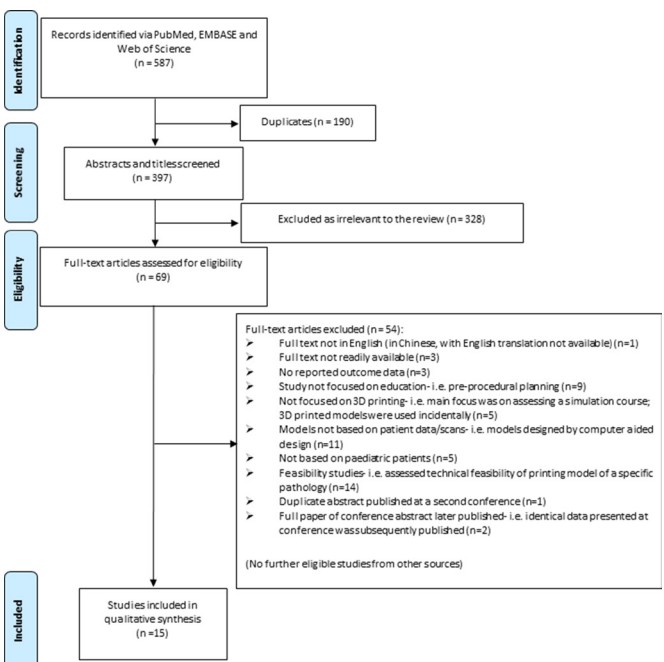

**Figure 1** Preferred Reporting Items for Systematic Reviews and Meta-Analyses flowchart of identifying eligible studies.

undergraduates.[13–15] One study was a multicentre study,[10] and only two studies followed up participants after a period of time.[13 16] The majority of studies were non-randomised, single-centred cross-sectional studies with no sample size calculations to determine the optimal number of study participants. Tables 1–3 summarise the study design, participants and findings of the included studies.

### Primary outcomes
Studies reported various objective and subjective (self-reported) outcome measures across a range of parameters, including knowledge acquisition, knowledge reporting, structural conceptualisation and identification, confidence in management and in explaining anatomy to others, learner satisfaction and increased subject interest. The key findings of each study are summarised in tables 1–3.

Objective assessment of 3DP models in the setting of CHD generally reported observable improvement in knowledge acquisition, structural conceptualisation and greater improvement from baseline assessment scores compared with control groups.[17–19] However, observable differences were not always statistically significant.[14 20 21] Those who used 3DP models reported improved subject understanding, increased confidence in reporting the pathology and in identifying the pathology compared with controls.[14 19–21] Participants felt that 3DP models were more useful than diagrams and videos and would like for these models to be implemented in their usual training.[15 22 23] However, from a surgical standpoint, while useful in training and demonstrating procedures, the textural properties of the material used did not match that of myocardial tissue.[10]

The studies in paediatric gynaecology and general surgery revealed better self-reported and objective measured outcomes when using 3DP models.[24–26]

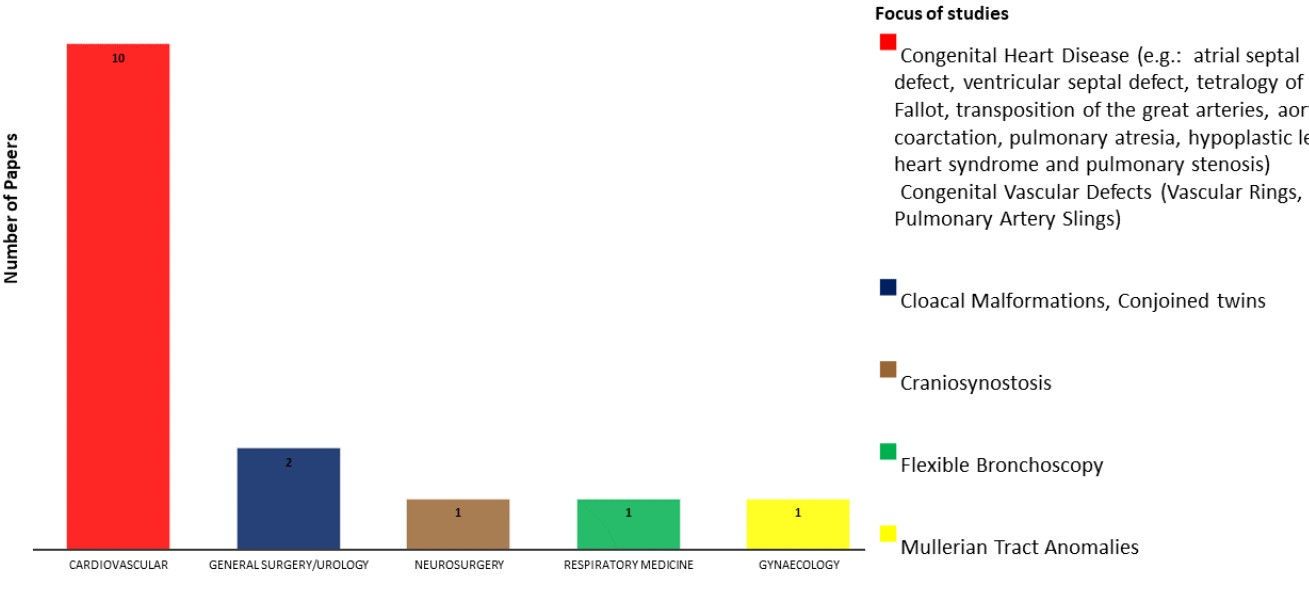

**Figure 2** Breakdown of specialties for included studies.

**Table 1** Study characteristics of cross-sectional studies

| Study | Design and objective | Study cohort | Primary outcomes | Secondary outcomes |
|---|---|---|---|---|
| Facilitating surgeon understanding of complex anatomy using a three-dimensional printed model.[24] | Use of 3DP to improve understanding of complex anatomy (conjoined twins) versus CT scan and digital reconstruction. | 21 (12 paediatric surgery attendings, 9 paediatric surgery and general surgery residents). | 3DP models improved scale and shape orientation and identification of anatomy, but not linear/point-to-point distances.<br>Not all outcome data were reported.<br>Using 3DP models was associated with improved test times compared with CT alone and CT with digital reconstructions (6.6 min vs 18.9 min vs 14.9 min, p<0.05). | None reported. |
| Hands-on surgical training of congenital heart surgery using 3-dimensional print models.[10] | Use of 3DP as part of surgical training for CHD. Participants were given a questionnaire following the surgical simulation session. | 81 cardiovascular surgery trainees and attendings Conducted in the USA, Canada and South Korea. | 3DP models demonstrated necessary pathological findings and were acceptable for surgical training. Materials used differed from real human myocardium. | Printing took 5–7 hours for each model at an estimated total cost of $150–$210. The average cost for print materials per model was $60. |
| Incorporating three-dimensional printing into a simulation-based congenital heart disease and critical care training curriculum for resident physicians.[17] | Use of 3DP models in a 60 min simulation teaching VSD anatomy, echocardiography, repair and postoperative critical care management Participants were given a presession and postsession test scored out of 10. | 23 paediatric resident physicians. | 3DP models improved their knowledge acquisition (4.83 vs 7.33, p=0.0082), knowledge reporting (4.25 vs 6.86, p=0.01) and structural conceptualisation (4.17 vs 7.22, p<0.0001). | None reported. |
| Transcending dimensions: a comparative analysis of cloaca imaging in advancing the surgeon's understanding of complex anatomy.[25] | To compare the effectiveness of four different modalities in teaching cloacal malformations in the context of operative planning (2D contrast study cloacagram vs 3D rotatable CT scan reconstruction vs software-enhanced 3D video animation vs 3DP cloaca model). | 59 paediatric surgeons (29 trainees and 30 attendings). | Participants using 3DP models scored significantly better (p<0.001) compared with those using other modalities.<br>2D cloacogram versus 3D CT versus enhanced 3D CT versus 3DP; trainees: 10.5% vs 46.7% vs 67.1% vs 73.8%; attendings: 22.2%, 54.8%, 66.2%, 74.0%. | 18 hours to print, cost of models not reported. |
| Use of 3D models of congenital heart disease as an education tool for cardiac nurses.[22] | Use of 3DP models to improve knowledge in cardiac nurses of various CHDs after treatment: TOF, transposition of the great arteries, aortic coarctation, pulmonary atresia, hypoplastic left heart syndrome Participants were given a five-question survey with Likert questions. | 100 cardiac nurses (65 paediatric, 35 adult). | Percentage of participants that agree/strongly agree that 3DP models:<br>► Improved their learning experience: 60%.<br>► Improved understanding of anatomy: 86%.<br>► Provided spatial orientation: 70%.<br>► Displayed anatomical complexity after treatment: 66%.<br>► Provided more information than diagrams: 74%.<br>Mean score for utility: 5.1 out of 7.0<br>Participants reported the value of being able to see inside the model heart explicitly and potential of using 3DP models in education/training. | None reported. |
| Utility of 3D printed cardiac models for medical student education in congenital heart disease: across a spectrum of disease severity.[15] | The impact of 3DP models for PS, ASD, aortic coarctation, TOF, TGA and HLHS. A four-station workshop used 2D images, embryology videos, spoken explanation, pathology specimens and 3DP models. Students rotated between all stations. They were given prestation and poststation questionnaires at the 3DP station assessing self-reported confidence and ranked the different teaching modalities. | 45 first-year medical students. | Using 3DP models improved self-reported confidence scores (PS (0.3, p<0.001), ASD (0.6, p<0.001), TOF (0.8, p<0.001), dextro-TGA (d-TGA) (0.9, p<0.001), Coarct (0.8, p<0.001), HLHS (1.1, p<0.001)).<br>Strong correlation between the complexity of pathology and perceived knowledge increase ($R^2$=0.73, p=0.03)<br>Participants agreed that they would use 3DP models in future teaching sessions (mean score: 4.40 out of 5).<br>74.2% of participants scored 3DP models at least 3 out of 5 for utility as an educational tool. | None reported. |
| Utility of 3-dimensional printing of hearts with complex congenital heart disease in the education of pediatric trainees and sonographers.* [23] | Use of 3DP models versus CT/MRI imaging to improve general understanding of CHD. | 10 (5 Paediatric residents, 2 sonographers and 2 cardiology fellows). | 80% of residents and 100% of sonographers and fellows felt that their understanding of CHD improved with 3DP models and felt that these should be used in the education of paediatric trainees. | None reported. |
| Utilizing three-dimensional printing technology to assess the feasibility of high-fidelity synthetic ventricular septal defect models for simulation in medical education.[18] | Teaching and simulation using 3DP CHD models, including instruction on surgical incisions and suturing skills Score out of 10. | 29 (16 medical and 13 premedical students). | Significant (p<0.0001) preseminar and postseminar score improvement in knowledge acquisition (3.22 vs 7.02), knowledge reporting (2.16 vs 6.60) and structural conceptualisation (2.17 vs 6.31) of VSD. | None reported. |

*Conference abstract.
ASD, atrial septal defect; CHD, congenital heart disease; 3DP, 3D printed; HLHS, hypoplastic left heart syndrome; PS, pulmonic stenosis; TGA, transposition of the great arteries; TOF, tetralogy of Fallot; VSD, ventricular septal defect.

**Table 2** Study characteristics of the cohort studies

| Study | Design and objective | Study cohort | Primary outcomes | Secondary outcomes |
|---|---|---|---|---|
| Assessing the utility of 3D printed models of Mullerian tract anomalies for clinical education.*[26] | Impact of 3DP models on trainees' understanding of Mullerian tract anomalies and assessment of the feasibility of producing such models. | Gynaecologists and general surgery trainees (exact number of participants not given). | 3DP models were found to increase gynaecologists' understanding of Mullerian tract anomalies and their confidence in surgery to correct them. | None reported. |
| Developing a 3D composite training model for cranial remodelling.[13] | Use of 3DP models at two annual practical courses to teach neurosurgical techniques (FOA and ES) for correction of craniosynostosis Surveys were given to attendees and non-attendees a year after the course. | 33 students, resident, fellows and attending surgeons over 2 years. | Models were a valuable training tool for surgical techniques and improved understanding and preparedness to perform the procedures. | None reported. |

*Conference abstract.
3DP, 3D printed; ES, endoscopic suturectomy; FOA, fronto-orbital advancement.

Identification of abdominal organs and cloacal malformations were more accurate and performed faster in those in the intervention arm of the studies compared with the control arm.[24 25] 3DP models left participants with greater self-reported confidence in understanding Mullerian tract anomalies.[26]

The two studies that followed up participants assessed the impact of 3DP models on bronchoscopy and neurosurgical training.[13 16] Those who performed bronchoscopy on 3DP models compared with those who did not perform assessment tasks, such as marker identification and completion time, were significantly better in immediate postsession testing. However, these benefits were not noticeable on follow-up a minimum of 2 months after the initial session.[16] Models produced for fronto-orbital advancement and endoscopic suturectomy simulation lead to improved understanding among attendees at the neurosurgical workshop, and at follow-up, participants felt that they were more prepared compared with their colleagues who did not attend.[13] This study, however, did not perform any quantitative analysis and did not objectively measure the impact 3DP models had at immediate or post-follow-up evaluation.

### Cost and duration of model generation

In four studies, some information was given about the cost or the time taken to generate the 3DP models.[10 16 20 25] Cost estimations varied widely, from $15 to $2500, depending on the type of printer, the material used and whether other costs were included. Segmentation/model production time was also included for two of the CHD studies. In one study, this was from 5 to 7 hours per CHD model to print, plus another 1 hour for cleaning and dyeing.[10] In another study, it took approximately 2 hours per tetralogy of Fallot model for segmentation and approximately 12 hours to print the model.[20] DeBoer et al claimed less than 12 hours for segmentation of an airway model.[16] Gasior et al reported that it took approximately 18 hours to print a 3D model demonstrating cloacal malformation.[25]

### DISCUSSION

The 15 studies included in this review demonstrate objectively measured and user-reported benefit in using 3DP models in paediatric trainee and medical student training and educational settings in a range of subspecialties, with a variety of outcomes. The quality of the evidence, however, is diminished by a number of factors.

There are limitations to our review and sources of bias within the evidence base. Many studies were single-centre, cross-sectional studies with a small number of study participants and no long-term follow-up, which produce lower-quality results according to validated tools of assessing medical educational studies.[27] The exception was one study which recruited participants from the USA, South Korea and Canada.[10] One study from our search was in Chinese, and an English translation of the full paper was not available—it was therefore excluded.[28] Due to the heterogeneity in outcome measures, we were not able to pool the results and directly compare the included studies, for example, as a meta-analysis.

A number of the studies did not take into account potential confounding factors. For instance, Smerling et al[15] measured the participants' self-reported understanding of CHDs immediately before and after exposure to 3DP CHD models. Each group entering the 3DP model station, however, had differing exposures, depending on which group they were randomised to. Those who ended the workshop with the 3DP station would have previously watched an embryology video, attended a lecture on CHDs and examined pathology stations, and hence may have reported greater confidence in understanding of CHDs compared with those who started the workshop at the 3DP station.

The lack of randomisation, coupled with studies not reporting how participants were recruited or randomised, is a source of selection bias. Some studies do not report baseline characteristics of participants,[13 18] which means that the impact of using 3DP models may

**Table 3** Study characteristics of randomised control trials

| Study | Design and objective | Study cohort | Primary outcomes | Secondary outcomes |
|---|---|---|---|---|
| Three-dimensional printed paediatric airway model improves novice learners' flexible bronchoscopy skills with minimal direct teaching from faculty.[16] | The impact of 3DP airway models in training/teaching flexible bronchoscopy to paediatric residents. Assessed on identifying six structures on bronchoscopy and the time taken to complete the task. Control and intervention groups were assessed prestudy, poststudy and a minimum of 2 months following the session. | 27 paediatric residents (PGY2) at the beginning of their paediatric respiratory rotation (18 in intervention group, 9 in control group). | Intervention versus control groups: median difference between prestudy and poststudy scores 4 vs 0 (p<0.001), median difference in times: 432 s vs 0 s (p<0.001). Minimum of 2 months poststudy, intervention versus control: median prestudy and delayed poststudy score difference (2.5 vs 1, p=0.123), median difference in prestudy and delayed poststudy times: 180 s vs 0 s (p=0.141). | Estimated total cost for materials and labour to trainer: $2500. Cost of fused deposition modelling-type 3D printer: $32 000. 12 hours to produce the computer model ready to print. |
| Three-dimensional printing models in congenital heart disease education for medical students: a controlled comparative study.[14] | To compare knowledge acquisition and structural conceptualisation of three subtypes of VSD for medical students with 3DP models versus without 3DP models. Participants were given a postsession test assessing knowledge acquisition and structural conceptualisation, and a subjective questionnaire. | 63 medical students (32 in the intervention group, 31 in the control group). | 3DP models significantly improved subjective understanding (mean score for intervention and control groups, out of 100: 72.19 vs 56.12; p<0.0001) and objective structural conceptualisation (mean score for intervention and control groups, out of 30: 18.44 vs 14.52; p=0.03) but not in knowledge acquisition of VSD (mean score for intervention and control groups, out of 70: 44.06 vs 36.77; p=0.06). | None reported. |
| Usage of 3D models of tetralogy of Fallot for medical education: impact on learning congenital heart disease.[20] | To compare conventional 2D drawings versus 3DP models in knowledge acquisition of TOF. Presession and postsession knowledge tests (out of 9) were performed, and self-reported questionnaires (out of 25) were completed. | 35 paediatric residents (17 in 2D image group, 18 in 3DP model group). | No observable significant difference between 3DP models and 2D images in terms of knowledge acquisition (mean post-test scores: 6.0 vs 6.3) or self-reported confidence in TOF (21 vs 20, p=0.39), but 3DP models provided significantly greater learner satisfaction (24 vs 21, p=0.03). | Cost to produce models ranged from $15 to $300 depending on the printer. Time taken to print each model was approximately 12 hours. |
| Use of 3D models of vascular rings and slings to improve resident education.[21] | Block randomisation was employed to recruit participants. The intervention group used 3DP models in didactic teaching sessions to improve understanding of vascular rings and pulmonary artery slings. The control group was shown virtual models. Presession and postsession subjective questionnaires and knowledge testing were performed by participants. | 36 paediatric and emergency medicine residents. | Both groups self-reported improved confidence in identifying, diagnosing, and treating vascular rings and slings. Intervention groups scored significantly higher than the control groups (62.2% vs 45.1%, p=0.001); however, the score improvement from pretest to post-test scores was insignificant between intervention and control groups (2.6 vs 1.8, p=0.084). | None reported. |
| Utility of three-dimensional models in resident education on simple and complex intracardiac congenital heart defects[19] | Block randomisation was employed to recruit participants. Both groups attended a lecture with 2D images/virtual models. The intervention group was given 3DP models during the lecture as an adjunct to the lecture content. Presession and postsession subjective questionnaires and knowledge testing were performed by participants. | 60 paediatric and emergency medicine residents (26 in the VSD portion of the study and 34 in the TOF portion of the study). | Subjective reporting of confidence in understanding the pathology significantly increased in both control and intervention groups in the VSD and TOF portions of the study. In the VSD study, the control group had a significantly greater improvement in post-test scores compared with those in the intervention arm (3.16 vs 1.93, p=0.004). In the TOF study, despite the intervention arm having a greater postsession test score than the control group (6.06 vs 5.29, p=0.037), there was no significant difference in presession and postsession score changes between control and intervention groups (2.23 vs 2.65, p=0.406). | None reported. |

3DP, 3D printed; TOF, tetralogy of Fallot; VSD, ventricular septal defect.

not be accurately reflected in outcome measurements. Cheng *et al*[13] analysed only subjective outcome measures such as confidence in understanding the pathology in focus; however, objective measures were not scrutinised. Another source of bias in this study arises from the questions posed to the control and intervention groups. Questions to the control group included 'I understand the anatomy…' and 'My training has fully prepared me for performing….' However, those in the intervention group were asked whether '…the model improved my understanding …' and 'the model was a valuable training tool….' Wording inconsistencies may cause the questions to become leading.

In an educational setting, the reporting of knowledge retention and translatability of knowledge and skill retention is key. Despite this, only one study reported long-term outcomes, which showed that there was no significant difference in the performance of participants on a testing apparatus for flexible bronchoscopy when followed up for a minimum of 2 months after the original training session.[16] No study assessed if training procedures with 3DP led to improved clinical outcomes. Outcome measures such as time taken to complete procedures in a clinical setting or procedural success/complication rates have been considered in studies focussing on simulation in medical education.[29 30] It has also

been demonstrated that using 3DP models for preoperative planning reduced operating theatre times by up to an hour.[31] There is potential for future paediatric studies to evaluate the clinical translation of 3DP model use in training.

Ten studies were based on CHD, possibly due to CHDs being the most common birth defect.[32] Traditional teaching of CHD has been delivered using 2D diagrams (eg, in textbooks), cross-sectional imaging modalities (MRI, CT and echocardiography), cadaveric specimens and off-the-shelf models.[33] However, these tools pose a variety of problems, including difficulty in translating planar images into 3D reconstructions in the learner's mind, lack of tactile feedback in the case of imaging, and lack of availability of specific pathologies in the case of cadaveric specimens and plastic models. Studies focussing on CHD suggest that 3D models can overcome these issues and have demonstrated increased understanding with more complex models.[15] Congenital defects can be complex concepts to grasp,[34] and therefore, 3DP models have the potential to be a very useful educational tool to implement into routine practice. This is especially helpful for adult clinicians who will be involved in transition-to-adult services to provide care for the growing population of adults with congenital diseases.[35 36] As CHD can be challenging for those with minimal training,[34] it would be beneficial for future studies to assess the use of 3DP models for teaching non-paediatric clinicians.

When producing models for procedure simulation or teaching, the shape and size of the model should be considered, along with the suitability of the material used. Acrylonitrile butadiene styrene was shown to most accurately mimic bone when used with a drill and for craniotomy.[13] However, replicating the consistency of soft tissue such as myocardium has proven difficult[10] and could detract from simulating the overall surgical experience. Another challenge posed is the lack of surrounding structures (eg, the dura) and the inability to replicate bleeds or leaks (eg, cerebrospinal fluid).[13] However, it has been demonstrated that soft tissues and artificial blood can be introduced into 3DP simulation models to replicate more closely in vivo anatomy and physiology.[37]

While here we focus specifically on paediatric teaching and training, other reviews exploring the use of 3DP models for undergraduate anatomy education and surgical training reported similar observations. Comparison of 3DP models to lectures, textbooks and 2D radiological imaging demonstrated a statistically significant improvement in medical student performance in anatomy tests.[38–40] As a tool for surgical trainees, participants using 3DP models reported greater confidence in understanding surgical pathologies and intraoperative confidence in performing procedures.[41 42] These systematic reviews also identified that many of these studies had non-randomised cohorts, small sample sizes and/or inconsistencies in measuring or reporting outcomes (such as differing pretest vs post-test formatting or omission of certain measurements). These limitations are consistent with our findings and should prompt improvements in the design of future educational studies using 3DP in order to systematically and robustly further understand the role of the technology in teaching and training.

We also reviewed the cost of model production. Paediatric trainees in low-income and middle-income countries (LMICs) can lack training opportunities due to limited traditional teaching resources.[3 43] 3DP models may have a role in improving access to training resources in such countries. DeBoer and colleagues highlighted that their $2500 3DP training model cost 43%–250% less than other commercially available trainers at the time of publication, with simple maintenance requirements.[16] Using 3DP medical aids has shown benefit for patients in Sierra Leone[44]; however, there is a paucity of evidence in the use of the technology in paediatric teaching and training in LMICs, which would warrant further investigation into its cost-effectiveness.

The current evidence base demonstrates a positive impact made by 3DP models on paediatric education and training—objectively and subjectively—for medical students and clinical trainees. The majority of studies failed to address sources of bias, assess clinically translatable outcomes or evaluate long-term benefits. Future studies should recruit larger cohorts, ideally at a multicentre level, randomise participants and adopt objective outcome measurements (eg, anonymous anatomy examinations and monitoring of trainee clinical performance), including follow-up to assess long-term benefits. 3DP models also have the potential to be more sophisticated to simulate real scenarios including soft tissue, surrounding structures and blood.

**Acknowledgements** The authors acknowledge the generous support of the British Heart Foundation (CH/17/1/32804) and The Grand Appeal (Bristol Children's Hospital Charity).

**Contributors** AA conceived the study and drafted the study protocol, registered the review to PROSPERO, conducted the review searches, was first reviewer of the eligible studies and led the writing of the manuscript. EL reviewed eligible papers and contributed to the writing of the manuscript. MC assisted in editing the final manuscript. GB assisted in editing the final manuscript. AS conceived and supervised the study, was adjudicator for any conflicts in study selection for the review, assisted in editing the final manuscript, and acts as guarantor of the study. All authors read and approved the final manuscript.

**Funding** The authors received no specific grant from any funding agency in the public, commercial or not-for-profit sectors for conducting this review.

**Competing interests** None declared.

**Patient consent for publication** Not required.

**Provenance and peer review** Not commissioned; externally peer reviewed.

**Data availability statement** Data are available upon reasonable request. All data relevant to the study are included in the article or uploaded as supplementary information. All eligible papers selected in the review are referenced in the final manuscript. Full PDF texts are available upon request.

**ORCID iDs**
Ashar Asif http://orcid.org/0000-0002-9361-1083
Elgin Lee http://orcid.org/0000-0002-0054-1574
Massimo Caputo http://orcid.org/0000-0001-7508-0891
Giovanni Biglino http://orcid.org/0000-0003-0413-149X
Andrew Ian Underwood Shearn http://orcid.org/0000-0002-3741-9118

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
