## [Reviewer comments · BMJ Paediatrics Open]

ARTICLE DETAILS

TITLE (PROVISIONAL)	The Role of 3D Printing Technology in Paediatric Teaching and Training: a Systematic Review
AUTHORS	Asif, Ashar Lee, Elgin Caputo, Massimo Biglino, Giovanni Shearn, Andrew

VERSION 1 – REVIEW

REVIEWER	Reviewer name: Dr. Nigel Drury Institution and Country: Birmingham Children's Hospital Paediatric Cardiac Surgery, United Kingdom of Great Britain and Northern Ireland Competing interests: None
REVIEW RETURNED	11-Feb-2021

GENERAL COMMENTS	This manuscript reports a systematic review on the role of 3D printed models of congenital disorders in the training of paediatricians, surgeons, medical students, and nurses. The authors searched standard bibliometric databases and from 397 abstracts screened, 15 studies were included. These were mostly observational studies but included 5 RCTs, and two thirds were focused on congenital heart defects. They identified some measurable benefits over traditional teaching aids and the 3D models were widely popular amongst participants. Where reported, the cost of producing the models varied hugely. The authors conclude that the evidence supporting potential benefit is weak and larger, better designed studies are needed. The systematic review was registered on Prospero prior to completion of the searches. The manuscript is fairly well written and the approach to the systematic review was very good. However, there are a few areas in which it would benefit from additional information or clarification: - Abstract, results (p5-6): Suggest including that 5/15 were RCTs and 10/15 were focused on congenital heart disease as this information would be useful up front.- Participants: This is well defined in the Prospero registration but is less clear in the manuscript. I suggest to define as 'paediatricians, surgeons, medical students, and nurses' in the abstract (background, p5) and provide greater detail in the selection criteria section (p8) similar to that in Prospero: 'Undergraduate medical students and clinical staff of all grades and professional backgrounds (including physicians, surgeons, nurses and allied health professionals) undergoing teaching, training or simulation in a field of paediatrics as part of an educational syllabus, training programme or professional course.'- Databases: The authors searched PubMed, Web of Science, EMBASE and Ovid Medline (p5,8), and of 640 records, 243 duplicates were identified and removed (figure 1). The high number of duplicates is not surprising as Medline, the US National Library of
--

	Medicine journal citation database, is the largest subset of databases searched by PubMed, the NLM's free online search engine. It therefore does not make sense to search both and presumably the Medline search found no additional studies to the PubMed search; as the larger resource, I suggest only including PubMed. - Literature search strategy (p5,8): The search terms are not adequately described in the manuscript to enable the searches to be repeated. The actual search terms should be included, as per the Prospero record, if these were the final terms used. The PRISMA checklist (p27, point 8) refers to the Prospero registration but readers should not be expected to look this up to understand how the search was conducted. Were MeSH headings used in PubMed/Medline? Where the search terms the same for all databases? If these were different and more extensive, could be put into an online only supplement. The Data Availability Statement (p26) refers to supplemental information but none was available for review. - Characteristics of selected studies (p10): This should be better described as 'specialities relating to paediatrics or congenital disorders'. - Assessment of bias: The Prospero record describes risk of bias (quality) assessments, but these were not included in the manuscript and in the PRISMA checklist (p28), they are described as N/A. Why were these not undertaken/reported? - Discussion: The discussion makes no reference to any systematic reviews outside of paediatrics and congenital disorders. Have others looked at the role in 3D models in teaching outside of paediatrics and if so, were similar issues identified? Any such papers should be briefly discussed and cited. - Table 1: The table is large and comprehensive but is the level of detail required in all areas? This study raises no apparent ethical issues.
--	---

REVIEWER	Reviewer name: Dr. Aurelio Secinaro Institution and Country: Ospedale Pediatrico Bambino Gesu, Italy Competing interests: None
REVIEW RETURNED	24-Feb-2021

GENERAL COMMENTS	Excellent Systematic Review. The very first one on this topic in children. I have non specific comment/suggestion.
---

VERSION 1 – AUTHOR RESPONSE

We would like to thank the Editor for considering our paper for publication and the peer reviewers for their thorough appraisal and useful comments to our review. Below are the comments from the Editor and peer reviewers, and our responses are highlighted below:

Divide Table 1 into several tables - I suggest one containing RCTs, one containing the cross-sectional studies and one cohort studies.

The tables have been divided into 3 tables according to their study design.

I note that one study was not in English - please state the language and explain why translation was not an option. Add as a limitation in the Discussion or better still include in your paper

This study was in Chinese, and a full text translation was not available. This is now mentioned in the discussion as a limitation.

- Abstract, results (p5-6): Suggest including that 5/15 were RCTs and 10/15 were focused on congenital heart disease as this information would be useful up front.

This has been added to the results subheading of the abstract.

- Participants: This is well defined in the Prospero registration but is less clear in the manuscript. I suggest to define as 'paediatricians, surgeons, medical students, and nurses' in the abstract (background, p5) and provide greater detail in the selection criteria section (p8) similar to that in Prospero: 'Undergraduate medical students and clinical staff of all grades and professional backgrounds (including physicians, surgeons, nurses and allied health professionals) undergoing teaching, training or simulation in a field of paediatrics as part of an educational syllabus, training programme or professional course.'

These suggestions have been taken on board and added to the manuscript to reflect the details used in the Prospero entry.

- Databases: The authors searched PubMed, Web of Science, EMBASE and Ovid Medline (p5,8), and of 640 records, 243 duplicates were identified and removed (figure 1). The high number of duplicates is not surprising as Medline, the US National Library of Medicine journal citation database, is the largest subset of databases searched by PubMed, the NLM's free online search engine. It therefore does not make sense to search both and presumably the Medline search found no additional studies to the PubMed search; as the larger resource, I suggest only including PubMed.

We have reviewed the results from our Medline and PubMed searches. All 53 results obtained in the Medline search were also obtained from the PubMed search, as you had raised. For that reason we will remove Medline from our included databases changing our total papers obtained from searching to 587 and number of duplicates to 190. This has now been changed in the manuscript and in figure 1.

- Literature search strategy (p5,8): The search terms are not adequately described in the manuscript to enable the searches to be repeated. The actual search terms should be included, as per the Prospero record, if these were the final terms used. The PRISMA checklist (p27, point 8) refers to the Prospero registration but readers should not be expected to look this up to understand how the search was conducted. Were MeSH headings used in PubMed/Medline? Where the search terms the same for all databases? If these were different and more extensive, could be put into an online only supplement. The Data Availability Statement (p26) refers to supplemental information but none was available for review.

The same search terms that are quoted in the Prospero entry were used across all databases. On PubMed, all searches included MeSH headings. This is now added in the methods section of the manuscript.

- Characteristics of selected studies (p10): This should be better described as 'specialities relating to paediatrics or congenital disorders'.

This has been updated in the manuscript.

- Assessment of bias: The Prospero record describes risk of bias (quality) assessments, but these were not included in the manuscript and in the PRISMA checklist (p28), they are described as N/A. Why were these not undertaken/reported?

Under "Risk of Bias Assessment", our Prospero record states "The quality of studies will be assessed according to the preferred reporting standards for each type of study: RCTs: CONSORT Statement, Observational studies: STROBE Statement, Case reports and series: CARE Statement." When appraising the eligible papers, we referred and checked against to the appropriate reporting standards for each study type. Within the discussion, we address sources of bias such as lack of randomisation/reporting of participant selection, failure to address confounding and dubious outcome measuring that we noted during our appraisals whilst using the relevant reporting standards. These sections have been reworded to make this clearer, and I hope this clarifies your query.

- Discussion: The discussion makes no reference to any systematic reviews outside of paediatrics and congenital disorders. Have others looked at the role in 3D models in teaching outside of paediatrics and if so, were similar issues identified? Any such papers should be briefly discussed and cited.

Other studies looking at the use of 3D printed models in anatomy education, surgical training and neurosurgical simulation have been published. These have been included in the discussion.

- Table 1: The table is large and comprehensive but is the level of detail required in all areas?

The table has had the excessive detail removed. The manuscript itself does not make reference to any numerical values quoted in the studies and meta-analysis was not appropriate due to outcome measure heterogeneity. We felt that quoting data from each study contextualises their outcomes, however we do accept that there is some information in that table which is not relevant to the scope of our systematic review and has been removed. The table has also been separated into their study designs as per the advice of the Editor-in-Chief